# Platelet-Derived miR-126-3p Directly Targets AKT2 and Exerts Anti-Tumor Effects in Breast Cancer Cells: Further Insights in Platelet-Cancer Interplay

**DOI:** 10.3390/ijms23105484

**Published:** 2022-05-13

**Authors:** Matteo Sibilano, Valentina Tullio, Gaspare Adorno, Isabella Savini, Valeria Gasperi, Maria Valeria Catani

**Affiliations:** 1Department of Experimental Medicine, University of Rome Tor Vergata, 00133 Rome, Italy; matteosibilano@libero.it (M.S.); valentinatullio.nu@gmail.com (V.T.); savini@uniroma2.it (I.S.); 2Department of Biomedicine and Prevention, University of Rome Tor Vergata, 00133 Rome, Italy; gaspare.adorno@ptvonline.it

**Keywords:** breast cancer, microvesicles, miR-126, PIK3/AKT signaling, platelets, polyunsaturated fatty acids

## Abstract

Among the surrounding cells influencing tumor biology, platelets are recognized as novel players as they release microvesicles (MVs) that, once delivered to cancer cells, modulate signaling pathways related to cell growth and dissemination. We have previously shown that physiological delivery of platelet MVs enriched in miR-126 exerted anti-tumor effects in different breast cancer (BC) cell lines. Here, we seek further insight by identifying AKT2 kinase as a novel miR-126-3p direct target, as assessed by bioinformatic analysis and validated by luciferase assay. Both ectopic expression and platelet MV-mediated delivery of miR-126-3p downregulated AKT2 expression, thus suppressing proliferating and invading properties, in either triple negative (BT549 cells) or less aggressive Luminal A (MCF-7 cells) BC subtypes. Accordingly, as shown by bioinformatic analysis, both high miR-126 and low AKT2 levels were associated with favorable long-term prognosis in BC patients. Our results, together with the literature data, indicate that miR-126-3p exerts suppressor activity by specifically targeting components of the PIK3/AKT signaling cascade. Therefore, management of platelet-derived MV production and selective delivery of miR-126-3p to tumor cells may represent a useful tool in multimodal therapeutic approaches in BC patients.

## 1. Introduction

Tumor cells and platelets influence each other, establishing a fine-tuned “platelet-cancer loop”; indeed, platelets have been shown to actively influence cancer cell behavior, but, in parallel, their physiology and phenotype can be profoundly impacted by tumor cells [1]. This crosstalk is difficult to interpret since platelets might exert either protective or deleterious roles in all steps of cancer progression, depending on the cancer type, microenvironment, and chemotherapy treatments [1]. As an example, in metastatic melanoma patients, high concentrations of 12- and 15-hydroxyeicosatraenoic acids (respectively produced by platelet 12-lipoxygenase and cyclooxygenase-1) have been shown to exert pro-malignant effects [2,3,4], and blocking cyclooxygenase-1 by low-dose aspirin is associated with a reduction in risk of several cancer types [5]. Conversely, serotonin, released by platelets in response to various stimuli, may exert inhibitory effects on tumor onset and progression [6,7].

Some platelet-triggered effects are mediated by bioactive mediators (growth factors, cytokines, inflammatory molecules, mRNAs, and miRNAs) present inside microvesicles (MVs), abundantly released upon activation. MV delivery into recipient cells, therefore, contributes to regulation of the tumor microenvironment and cell-to-cell interactions [8,9,10]. For instance, cancer-promoting effects of MVs have been described due to their content of growth factors, inflammatory cytokines, and angiogenic mediators [11,12,13]. On the other hand, a tumor-suppressive role has also been put forward for their ability to deliver specific miRNAs, small, evolutionary conserved, single-stranded, non-coding RNAs, that modulate the expression of a plethora of genes regulating key biological events (including apoptosis, differentiation, development, proliferation, metabolism, and signal transduction) [14,15,16,17]. These phenomena may be explained by considering the recent literature data, which suggest that platelet-derived MVs may exert tumor-suppressive roles at early stages while supporting cancer progression and metastatic dissemination at late stages [1].

From its discovery and the elucidation of the role it plays in vascular integrity, several studies have pointed out the role of miR-126, one of the most thoroughly studied miRNAs in relation to platelet activity, and in cancer onset and progression. This miRNA, encoded by *hsa-mir-126* gene located in intron 7 of the *epidermal growth factor-like domain 7 (EGFL7)* gene [18,19], is usually downregulated in different tumors, including breast, lung, colon, bladder, gastric, esophageal, cervix, prostate, liver, and prostate cancers [20,21,22,23,24,25,26]. miR-126 exerts a tumor suppressor activity, regulating signaling pathways involved in tumorigenesis. Just as an example, miR-126 inhibits cell cycle progression from G1/G0 to S phase of breast cancer (BC) cells by directly targeting the insulin receptor substrate 1 (IRS-1) constitutively activated in tumors [27], while it reduces trastuzumab resistance of BC cells by directly targeting phosphoinositide-3-kinase regulatory subunit 2 (PIK3R2, also known as p85β) [28]. Accordingly, miR-126 loss and the subsequent increase in PIK3R2 provide a selective growth advantage in primary colon tumors [22]. A role for miR-126 in tumor neo-angiogenesis has also been postulated in lung carcinoma by Liu’s group [29], who demonstrated that miR-126 restoration downregulates vascular endothelial growth factor (VEGF), with subsequent cell cycle G1 arrest and reduction in tumor nodule formation in nude mice. Finally, loss of miR-126 in cancer cells may have profound effects on metastatic potential: by array-based miRNA profiling, miR-126 has been shown to be consistently downregulated in metastatic foci of BC and significantly associated with poor metastasis-free survival [30,31]. In this context, we have previously demonstrated that treatment of platelets with the ω6-polyunsaturated fatty acid (PUFA) arachidonic acid (AA) and the ω3-PUFA docosahexaenoic acid (DHA) significantly enhanced MV release, as well as miR-126 amounts inside [15]. Our data also showed that miR-126-3p-enriched platelet MVs were internalized by BC cells with relevant biological implications, such as blocking of the cell cycle, reduction in migration, and increased sensitivity to the chemotherapy drug cisplatin [15].

Among signaling pathways strictly associated with BC, the PIK3/AKT transduction cascade is crucially implicated in tumorigenesis. In particular, an isoform-specific AKT signaling has been shown to modulate several BC hallmarks, including cell growth, survival, and invasiveness [32]. Although highly homologous, the AKT isoforms exert, indeed, distinct, non-redundant effects in BC, with AKT1 displaying a tumor-initiating role and AKT2 being mainly involved in tumor progression and metastasis [32,33,34]. Despite conflicting findings, what emerged from different investigations is that AKT1 and AKT2 have somehow opposite actions on BC cells. AKT1, indeed, primarily controls cell proliferation and death, by acting on cell cycle (p21, p27, and cyclin D1) and apoptotic (p53) proteins, while inhibiting migration [34,35,36]; on the contrary, AKT2 notably promotes migration and invasion by regulating adhesion and cytoskeletal proteins, such as β-integrins, F-actin, and epithelial to mesenchymal transition (EMT) proteins [37,38,39]. According to its crucial role, several elements of the PIK3/AKT pathway are usually deregulated in BC [40,41,42], and AKT upregulation is a typical marker of a poor prognosis [42,43,44,45,46].

Given the known crosstalk between platelets and cancer cells [12,47], and based on the above described findings, we investigated the molecular mechanisms underlying miR-126-dependent regulation of platelet–BC cell crosstalk. To this aim we checked potential miR-126-3p molecular targets in BC cells that may account for the observed anti-tumor effects.

## 2. Results

### 2.1. Platelet-Derived miR-126 Directly Targets AKT2

We have previously shown that the ω6-PUFA AA (alone or in combination with the ω3-PUFA DHA) significantly increased miR-126 amounts inside platelet-derived MVs that, once internalized by BC cells, exerted an overall anti-cancer effect [15]. To gain further insight, we checked potential molecular targets of miR-126-3p, focusing on the PIK3/AKT pathway, a signaling cascade crucial for BC survival and usually deregulated in breast tumors [40,41,42]. By using the Target Scan 8.0 software, we identified AKT2 as a putative miR-126-3p target. To experimentally validate the bioinformatic data, we cloned the wild type and mutated *AKT2* 3′UTR (Figure 1a) into the pGL3-Control Luciferase Reporter Vector and carried out co-transfection experiments. In miR-126-3p co-transfected BT549 cells, the luciferase activity of the reporter vector containing the wild type *AKT2* 3′UTR significantly decreased with respect to scramble-transfected cells; conversely, in cells co-transfected with anti-mirR-126-3p, the luciferase activity was unchanged with respect to scramble-transfected cells (Figure 1b). Accordingly, “seed sequence”-based mutation of miR-126-3p binding site of 3′ UTR restored the luciferase activity, thus indicating a direct interaction of the *AKT2* 3′UTR with miR-126-3p (Figure 1b). In agreement with the results of the luciferase assay, ectopic miR-126-3p significantly downregulated protein expression of AKT2, as well as of PIK3R2, a well-recognized miR-126-3p target belonging to the PIK3/AKT signaling cascade [28] (Figure 1c); as expected, miR-126-3p inhibitor exerted opposite effects, leading to unchanged or increased levels of target proteins. This confirmed that an inverse relationship between the expression of miR-126-3p and AKT2 and PIK3R2 occurs in BT549 cells.

### 2.2. miR-126-3p-Mediated Downregulation of AKT2 Inhibits Migration and Invasiveness of BC Cells

To assess whether AKT2 might be directly involved in miR-126-3p-triggered anti-cancer effects, we performed a wound healing scratch assay in BT549 cells transiently transfected either with miR-126-3p mimic or a small interfering RNA (siRNA), selected among four commercially available siRNAs targeting AKT2 (Appendix A).

Either forced miR-126-3p expression or silencing of its target *AKT2* triggered identical effects (Figure 2). As early as 15 h after transfection, scramble-transfected cells fully covered the scratched area, reaching 100% confluence (Figure 2a,b); conversely, at the same incubation times, cells transfected with miR-126-3p (Figure 2a,b) or AKT2 siRNA (Figure 2a,b) showed delayed re-epithelialization of the wound area (40–50% of migration). Identical effects were observed in the less aggressive, luminal A subtype, MCF-7 cell line (data not shown), thus suggesting a more generalized action on other BC subtypes.

Accordingly, in the transwell migration assay, miR-126-3p-transfected cells displayed 52.3% inhibition of the migratory potential, while silencing of *AKT2* led to 98% inhibition (Figure 3a,c). Conversely, miR-126-inhibitor enhanced the ability of BC cells to invade in Matrigel-coated transwells (Figure 3a,c). Again, this effect was also observed in MCF-7 cells: both miR-126-3p mimic and AKT2 siRNA reduced the number of invading cells (89% and 78% of inhibition, respectively), while miR-126-3p inhibitor showed an opposite behavior (Figure 3b,c). Altogether, our data demonstrated that miR-126-3p/AKT2 axis is a strong suppressor of BC cell invasion.

### 2.3. miR-126-3p-Mediated Downregulation of AKT2 Inhibits Phosphorylation-Dependent Cofilin Activity

To dissect the potential mechanism(s) through which miR-126-3p inhibited cell migration, we investigated regulation of actin dynamics at the plasma membrane, especially focusing on cofilin activity, whose upregulation has been found to significantly associate with poorer outcome in BC [48]. As the severing and depolymerization activity of cofilin is dependent on phosphorylation/dephosphorylation cycles [49], we analyzed cofilin phosphorylation at Ser3, that abolishes the actin-binding activity, thus inactivating cofilin. As shown in Figure 4, phosphorylation levels significantly increased in miR-126-3p-transfected cells. Notably, we also found decreased cofilin protein levels, suggesting more than one point of regulation. Transient silencing of *AKT2* led to a similar phenotype (Figure 4), thus further indicating a connection between miR-126 and AKT2 in the regulation of cofilin activity.

### 2.4. miR-126-3p-Mediated Downregulation of AKT2 Decreases the Clonogenic Potential of BC Cells

We then assessed if miR-126 also affected the proliferating capacity of BC cells and if this effect might also involve AKT2. With a comparison to scramble-transfected cells, miR-126-3p-transfected cells showed reduced clonogenic potential, in both BT549 and MCF-7 cells. Indeed, miR-126-3p mimic and AKT2 siRNA significantly decreased the number and size of colonies with respect to scramble oligo; conversely, more and larger colonies were observed in miR-126-3p inhibitor-transfected samples (Figure 5).

### 2.5. Platelets Regulate BC Cell Behaviours via miR-126-3p-Mediated Downregulation of AKT2

We lastly assessed if the miR-126-3p/AKT2 pathway could also be functional during platelet–BC cell crosstalk. Therefore, we checked the effects of physiological delivery of miR-126 (*via* platelet MVs) into BC cells. According to our previous study [15], platelets activated with AA plus DHA released miR-126-enriched MVs (Figure 6a) that were internalized by BT549 cells with a subsequent increase in cellular miR-126 content (Figure 6b). Consequently, a significant drop in AKT2 and PIK3R2 protein levels was found (Figure 6c), paralleled by an increased amount of phosphorylated (inactive) and decreased expression of total cofilin (Figure 6c), and an 83% reduction of migratory potential (Figure 6d). Finally, once they had internalized miR-126-enriched MVs, BT549 cells developed colonies smaller and significantly farther distanced from each other than colonies formed by cells receiving MVs from resting platelets (Figure 6e).

### 2.6. Prognostic Value of miR-126 and AKT2 Levels in BC

Finally, to further validate our results in vivo, we performed an exploratory bioinformatic analysis of the relationship between miR-126 and AKT2 expression in BC. To this aim, by using the online Kaplan–Meier plotter database [50,51], we investigated the prognostic value of miR-126 on 5-year overall survival of BC subjects. As shown in Figure 7a, a significant positive correlation between miR-126 levels and overall survival in tumor specimens included in the METABRIC dataset was found; moreover, except for HER2 positive tumors (exhibiting only a trend), such a correlation was retained in all BC subtypes, with Luminal B and TNBC subtypes being the most significantly associated with unfavorable survival [HR = 0.59 (0.39–0.91), *p* = 0.015 and HR = 0.45 (0.23–0.89, *p* = 0.018, respectively] (Figure 7b–e); in addition, as emerged from the analysis of AKT2 protein levels [52,53], high AKT2 expression predicted a worse 5-year overall survival compared with low AKT2 expression [HR = 2.26 (1.01–5.09); *p* = 0.042] (Figure 7f). A lack of data relative to BC subtypes did not allow us to refine in detail AKT2 analysis.

These findings suggested that a potential inverse relationship between miR-126 and AKT2 occurs in BC patients and that miR-126 acts as a tumor suppressor by targeting, among others, AKT2.

## 3. Discussion

We previously demonstrated a positive role played by platelets in a host’s defense against BC, via MVs-mediated delivery of miR-126 [15]. In the present study, we deepened the molecular mechanism underlying the anti-cancer effects of platelet-derived miR-126. We revealed that AKT2 kinase is an miR-126-3p direct target and the miR-126-3p/AKT2 axis plays a crucial role in platelet/BC crosstalk. Indeed, the increase in miR-126 content in BC cells (*via* platelet MV delivery) led to the inhibition of AKT2 expression, thus suppressing the invading potential of tumor cells. These effects might be related to the inhibition of migration, subsequent to the accumulation of inactive phospho-cofilin. By stimulating the turnover of actin filaments at the protruding ends (and, therefore, controlling the direction of cell movement), cofilin is, indeed, a major determinant of migration, and phosphorylation status and activity of this protein are directly associated with the metastatic phenotype of BC cells [54,55,56,57]. Notably, we demonstrated that the miR126-3p/AKT2 axis significantly suppressed colony forming efficiency; this finding suggests that, similarly to the AKT1 isoform, AKT2 is also implicated in promoting cell proliferation, thus revealing a novel role for this kinase beyond the commitment to invasion and metastasis [32].

All events triggered by miR-126-3p-mediated downregulation of AKT2 seem to be functional in both receptor positive and negative BC cells, as superimposable effects were found in TNBC subtype (BT549 cells) and in a less aggressive Luminal A (MCF-7 cells) subtype. In this context, it should be stressed that the PIK3/AKT/mTOR signaling cascade is strictly associated with BC [58,59], and many of the genes encoding for elements of this pathway are mutated in this cancer type. Just as an example, up to 27% of BC patients have mutations in *PIK3CA*, the gene encoding the p110α subunit, that are found in luminal A (45%), HER2-enriched (39%), luminal B (30%), and TNBC (9%) subtypes [60,61]. Other alterations concern the constitutive activation of AKT (mutations occurring in 2% of BC patients) that is associated with a worse outcome and chemotherapy resistance [43,59,62]. Interestingly, PIK3/AKT signaling cascade is specifically targeted by the onco-suppressor miR-126, as it is well documented that *IRS-1* and *PIK3R2*, two upstream components of this pathway, are directly inhibited by miR-126 in several cancer types, including BC [22,27,28]. Here, we added a new piece to the complex miR-126 puzzle, by demonstrating specific downregulation of another component of this cascade, namely *AKT2*, thus further strengthening the hypothesis that miR-126-3p tumor suppressor activity relies on the inhibition of PIK3/AKT signaling (Figure 8).

The miR-126-3p/AKT2 axis established following the interaction between platelets and cancer cells could have clinical implications. As suggested by our bioinformatics data, either high miR-126 or low AKT2 levels predicted better outcomes of BC patients, as they correlated with 5-year overall survival. In particular, the prognostic value of miR-126 was retained in all BC subtypes, with Luminal B and TNBC subtypes being the most significantly associated. Therefore, it is conceivable that platelet MV-dependent delivery of miR-126 may represent a protective mechanism to cope with low miR-126 expression in BC. Circulating miR-126, hence, might represent a useful non-invasive biomarker for BC diagnosis and prognosis, as has already emerged for other specific miRNAs [63,64,65].

It is well established that platelets actively modulate all steps of cancer, with consequences strictly dependent on cancer type and MV composition. For example, platelet MVs infiltrating lung and colon cancers deliver miR-24 that leads to mitochondrial dysfunction and apoptotic cell death [16]; conversely, platelet MV-mediated transfer of miR-939 to ovarian cancer cells triggers epithelial to mesenchymal transition [66]. Accordingly, a cancer type-specific correlation can also be found between platelet count and cancer risk and/or clinical outcomes. Indeed, a high platelet number is frequently diagnosed as comorbidity and positively correlated with poor overall, progression-free, and recurrence-free survival in cervical, ovarian, lung, stomach, and colon tumors [67,68,69,70]. Conversely, a positive correlation between thrombocytosis and poor overall survival occurs in inflammatory, but not in non-inflammatory BC [71], and even thrombocytopenia at diagnosis has recently been reported to correlate with poor clinical outcomes in BC patients [72]. 

In conclusion, our study broadens the complex role played by platelets in both early and late stages of tumor progression. Further studies are clearly warranted as both tumor-suppressive and promoting effects have been reported for miR-126, most likely dependent on the oncogenic context. For example, high miR-126 levels promote in vivo leukemogenesis and correlate with poor prognosis in acute myeloid leukemia patients [73,74]. This apparent functional divergence of miR-126 action in solid and non-solid tumors needs to be further investigated in order to make use of miRNA targeting as a therapeutic approach for cancer patients. In this context, platelet horizontal miR-126-3p transfer and subsequent downregulation of PIK3/AKT2 signaling may be clinically relevant, especially for hypothesizing the future management of MV production and selective delivery to tumor cells.

## 4. Materials and Methods

### 4.1. Cell Cultures

Human triple negative breast cancer (TNBC) BT549 cells (ATCC HTB-122) were maintained in RPMI-1640 culture medium supplemented with 0.023 U/mL insulin; luminal A subtype MCF-7 cells (ATCC HTB-22) were grown in Dulbecco’s modified Eagle’s medium (DMEM). Both culture media were supplemented with 10% fetal bovine serum (FBS), 100 μg/mL kanamycin, 0.1 mg/mL sodium pyruvate (Biowest, Round Rock, TX, USA), and cells were cultured in a humidified 5% CO_2_ atmosphere at 37 °C.

During the study, cells were regularly tested for the absence of mycoplasma by using either MycoStrip™-Mycoplasma Detection Kit (InvivoGen, Toulouse, France) or LookOut^®^ Mycoplasma qPCR Detection Kit (Sigma Aldrich, Sant Luis, MI, USA).

### 4.2. Platelet Isolation

After informed consent (Prot. 685 12 January 2017; Independent Ethics Committee Policlinico Tor Vergata DDG n.546 12 August 2016), platelets were collected from fifteen healthy plateletpheresis donors of the Transfusion Center, Policlinico Tor Vergata, University of Rome; the remnants (1–2 mL) from samples used for routine platelet count were used. After centrifugation at 1000 × *g* for 15 min, the platelet pellet was resuspended in 5:1 (*v*/*v*) Tyrode (100 mM HEPES, 1.3 M NaCl, 29 mM KCl, 10 mM NaHCO_3_):ACD (80 mM glucose, 25 mM citric acid, 45 mM sodium citrate), containing 25 mM glucose and 2.5 mM CaCl_2_; the final platelet concentration was 1 × 10^8^/mL.

Platelet activation was carried out with 1 mM AA plus 500 µM DHA. PUFAs were dissolved in absolute ethanol, at a concentration 100-fold higher than working solution; the same amount of ethanol was added to control samples. Activation was carried out at 37 °C for 30 min with slight agitation, as already reported [15].

### 4.3. Isolation of Platelet-Derived MVs and Delivery to Cancer Cells

Both resting and activated platelets were centrifuged at 2000× *g* for 15 min at 4 °C, and the platelet MV-enriched plasma was further centrifuged at 100,000× *g* for 1 h at 4 °C in an MLS-50 ultracentrifuge (Beckman Coulter, Indianapolis, IN, USA). MVs were collected and either snap frozen for RNA extraction or resuspended in culture medium.

For delivery experiments, 1.5 × 10^5^ BT549 cells were seeded on 6-well plates the day before the experiment. Platelet-derived MVs were added to culture medium and incubated at 37 °C for 24 h, before collecting cells for further analyses, as described [15].

### 4.4. Bioinformatic Studies

The online accessible TargetScan Human 8.0 software was used to search and analyze potential miR-126-3p targets [75].

Clinical outcome analysis of miR-126 and AKT2 expression in BC patients was performed by Kaplan–Meier plotter database. Gene expression data and survival information of 1262 BC patients download from METABRIC were analyzed for *miR-126* [50,51], and protein expression data and survival information of 65 BC patients downloaded from [53] were analyzed for AKT2. Overall survival analysis was performed in settings with 60 months follow-up.

### 4.5. In Vitro Transient Transfections

BT549 and MCF-7 cells were transfected with 25 nM of either miR-126-3p-mimic (cod. AM17100, Thermo Fisher Scientific, Waltham, MA, USA), or miR-126 inhibitor (anti-miR-126, cod. AM17000, Thermo Fisher Scientific) or scramble negative control (cod. AM17110, Thermo Fisher Scientific), by using Lipofectamine RNAiMAX (Invitrogen, Heidelberg, Germany), according to manufacturer’s instructions. After 24 h, cells were harvested and/or processed for further experiments.

For *AKT2* silencing, BT549 and MCF-7 cells were transfected with 25 nM of AKT2 siRNA (Hs AKT2_8 cod. SI00287679; Qiagen, Germantown, MD, USA) or scramble negative control (cod. 1027281, Qiagen) by using Lipofectamine RNAiMAX (Invitrogen), according to manufacturer’s instructions. After 48 h, cells were collected and/or processed for further analyses.

### 4.6. Cloning of AKT2 3′UTR and Luciferase Assay

pGL3-3′UTR AKT2 plasmid was built as follows. A 997-bp fragment from human *AKT2* 3′UTR (Figure 1) was amplified with forward primer 5′-GGCCTCTAGAGTCTTTTTCCTCTGTGTGCGATG-3′ and reverse primer 5′-GGCCTCTAGAAACCCCAACCAAACGAGTCC-3′, and then ligated to pGL3-Control Luciferase Reporter Vector (Promega, Madison, WI, USA). The mutated putative miR-126-3p binding site in the *AKT2* 3′UTR was generated using the Quick-change site-directed mutagenesis kit (Agilent, Santa Clara, CA, USA), according to the manufacturer’s protocol.

For luciferase assay, 1.5 × 10^5^ BT549 cells were seeded into 12-well plates the day before transfection. Cells were transfected with either miR-126-3p-mimic, or anti-miR-126 or scramble negative control, 30 ng Renilla plasmid (Promoterless Renilla Luciferase Basic Vector E6881, Promega) and 3 µg pGL3-3′UTR *AKT2* (wild type or mutant), by using Lipofectamine 2000 (Invitrogen), according to manufacturer’s instructions. Twenty-four hours after transfection, luciferase activities of cellular extracts were measured by using a Dual Luciferase Reporter Assay System (Promega); light emission was measured over 10 s using the OPTOCOMP I luminometer (MGM Instruments, Inc., Hamden, CT, USA). Efficiency of transfection was normalized with the Renilla luciferase activity.

### 4.7. Scratch Wound Healing Assay

BT549 (1.0 × 10^4^) cells were seeded in 96-well plates the day before the experiment. Scratch test was carried out on cells transfected either with miR-126-3p-mimic or AKT2 siRNA, on Incucyte S3 (Essen Bioscience, Ann Arbor, MI, USA). Before scratching, samples were incubated with 10 µg/mL mitomycin C (Sigma Aldrich), at 37 °C for 1 h. Then, cross was scratched in the middle of the bottom of 96-well plates by using the Incucyte WoundMaker device, and cells were cultured at 37 °C for different incubation times; pictures were taken at the point of intersection of the scratch, for up to 24 h, by light microscope (Nikon, Amsterdam, The Netherlands).

### 4.8. Invasion Assay

Transiently transfected BT549 and MCF-7 cells were trypsinized, resuspended in serum-free medium, and seeded (1 × 10^5^ cells in 250 μL) into migration chambers (8 μm membrane pore sizes, BD Biosciences, Franklin Lakes, NJ, USA), previously coated with 50 μL Matrigel matrix (BD Biosciences; stock solution: 0.3 mg/mL). Chambers were then inserted into wells (24-well plate), containing 800 μL of medium supplemented with 20% FBS, and incubated at 37 °C for 24 (BT549 cells) or 48 (MCF-7 cells) hours. Non-invading cells (remaining on the upper surface of the membrane) were gently removed with a cotton swab, while invading cells (cells adhered to the lower surface) were fixed and stained with 6% glutaraldehyde, 0.5% crystal violet solution for 30 min. After washes with distilled water, membranes were dried, removed with a surgical blade, and mounted on glass slides. Pictures were taken with Leyca microscope at 2.5 X magnification and cells counted with ImageJ software (ImageJ, NIH).

### 4.9. Colony Forming Unit (CFU) Assay

Transiently transfected BT549 and MCF-7 cells were trypsinized, seeded in 6-well plates (100–500 cells/well), and grown at 37 °C for at least 14 days. Afterwards, colonies were fixed and stained with 6% glutaraldehyde, 0.5% crystal violet solution, for 30 min. After washes with distilled water, number of colonies was counted using an inverted phase contrast microscope (Zeiss; 10 X objective).

### 4.10. Real-Time PCR

Total RNA was extracted by using the mirVANA^TM^ miRNA isolation kit (Thermo Fisher Scientific), according to manufacturer’ instructions. Ten ng of total RNA were reverse transcribed with the TaqMan^®^ MicroRNA Reverse Transcription Kit (Thermo Fisher Scientific) and reverse transcription primers specific for *miR-126* (Thermo Fisher Scientific). The reaction was performed on Applied Biosystems at 16 °C for 30 min, 42 °C for 30 min, 85 °C for 5 min. The cDNA was then amplified on a 7500 Fast Real-Time PCR system (Thermo Fisher Scientific), by using the TaqMan Universal PCR master mix (Thermo Fisher Scientific) with processing at 95 °C for 10 min, 95 °C for 15 s, and 60 °C for 1 min, for 40 cycles. miRNA expression was calculated using the 2^−ΔΔCt^ method, after normalization with *RPL21*.

### 4.11. Western Blot

Protein samples (20–40 µg/lane) from whole lysates were subjected to SDS-PAGE under reducing conditions, electroblotted onto PVDF membranes, incubated with specific antibodies and detected with enhanced chemiluminescence kit (Santa Cruz Biotechnology, Santa Cruz, CA, USA). The specific primary antibodies used were anti-PIK3R2 (1:1000; cod. ab180967, Abcam, Cambridge, England), anti-AKT2 (1:500; cod. 2964S, Cell Signaling Technology, Danvers, MA, USA), anti-cofilin (1:1000; cod. CP1131, ECM Biosciences, Versailles, KY, USA), anti-p-cofilin (Ser3) (1:1000; cod. 3311S, Cell Signaling Technology), and anti-GAPDH (1:10000; cod. ab8245, Abcam).

### 4.12. Statistical Analysis

Data are expressed as the mean ± SEM of at least three experiments. Statistical evaluation was performed using Student’s *t*-test for unpaired data, one-way analysis of variance (ANOVA) followed by a Bonferroni *t* test, or ANOVA for repeated measures where appropriate. Differences were considered statistically significant at *p* < 0.05.

## Figures and Tables

**Figure 1 ijms-23-05484-f001:**
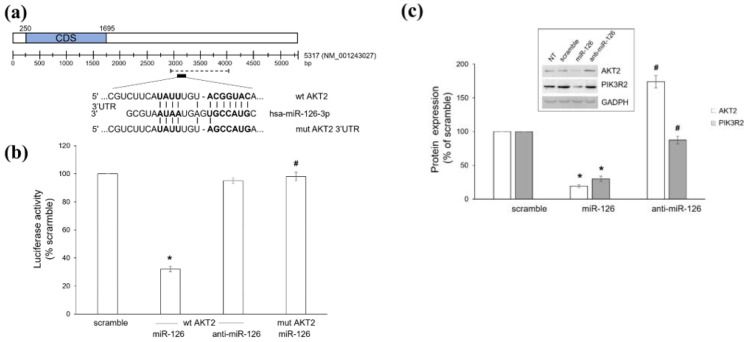
Ectopic miR-126-3p downregulates AKT2. (**a**) Predicted miR-126-3p target site on *AKT2* 3′UTR, as recognized by TargetScan Human 8.0 software. The sequence of wild type (wt) and mutant (mut) *AKT2* 3′UTR is shown; bold letters indicate the miR-126-3p seed sequence. Region cloned into the pGL3-Control Luciferase Reporter Vector is also indicated (dashed line). (**b**) Dual luciferase assay of wild type (wt) and mutated (mut) *AKT2* 3′UTR after transfection with scramble oligo (scramble), miR-126-3p-mimic (miR-126), or inhibitor (anti-miR-126). Values are reported as percentage of scramble-transfected cells, arbitrarily set to 100%. Results are shown as mean ± S.E.M. of four different measures from three independent experiments. * *p* < 0.01 versus scramble, # *p* < 0.01 versus wt AKT2 3′UTR. (**c**) Western blot of AKT2 and PIK3R2 in BT549 cells left untreated (NT) or transfected with either scramble oligo (scramble) or miR-126-3p mimic (miR-126) or inhibitor (anti-miR-126). Values are reported as percentage of scramble-transfected cells (not statistically differing from NT cells), arbitrarily set to 100%. Blots are representative of five independent experiments and results are shown as mean ± S.E.M. * *p* < 0.01 versus scramble, # *p* < 0.01 versus miR-126.

**Figure 2 ijms-23-05484-f002:**
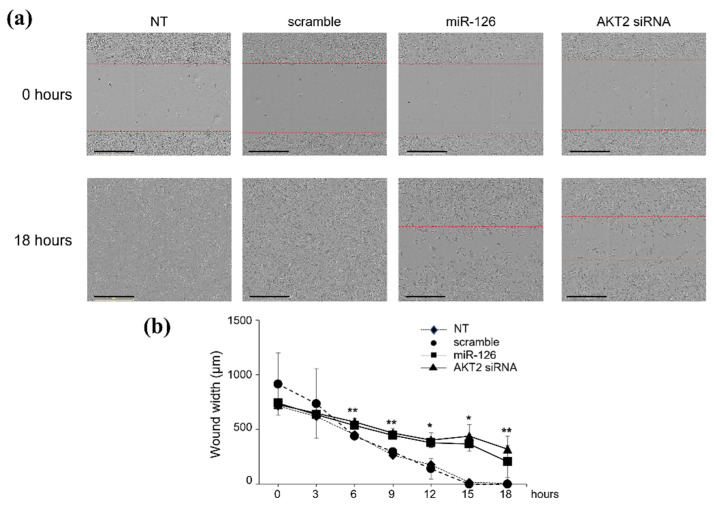
miR-126-3p/AKT2 axis reduces migration of BT549 cells. (**a**) Scratch wound assay performed on BT549 cells left untreated (NT) or transiently transfected with either scramble oligo (scramble) or miR-126-3p-mimic (miR-126) or AKT2 siRNA. Migration was monitored for up to 18 h. The leading edge of the wound is highlighted red. (**b**) Graphical view showing the wound width (µm), in relation to incubation times. Results are representative of three independent experiments, each performed in quintuplicate and reported as mean ± S.E.M. * *p* < 0.01 and ** *p* < 0.05 versus scramble. Scale bars: 400 µM.

**Figure 3 ijms-23-05484-f003:**
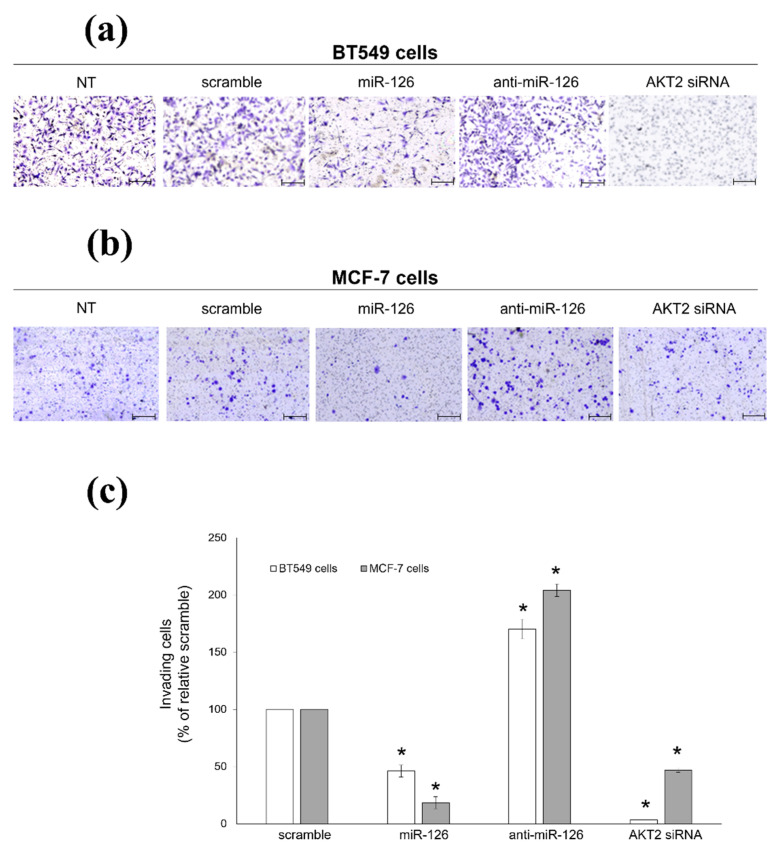
miR-126-3p/AKT2 axis reduces invasion of BC cells. Transwell migration assay performed with BT549 (**a**), and MCF-7 (**b**) cells left untreated (NT) or transiently transfected with either scramble oligo (scramble), miR-126-3p-mimic (miR-126), miR-126-3p-inhibitor (anti-miR-126), or AKT2 siRNA. Photographs are representative of four independent experiments. (**c**) Histograms show the number of invading cells, reported as percentage of relative scramble-transfected cells (not statistically differing from NT cells), arbitrarily set to 100% (absolute cell number = 608.12 ± 21.72 and 161.62 ± 14.67 for BT549 and MCF-7 cells, respectively). Data are shown as mean ± SEM. *: *p* < 0.01 versus relative scramble. Scale bars: 250 µm.

**Figure 4 ijms-23-05484-f004:**
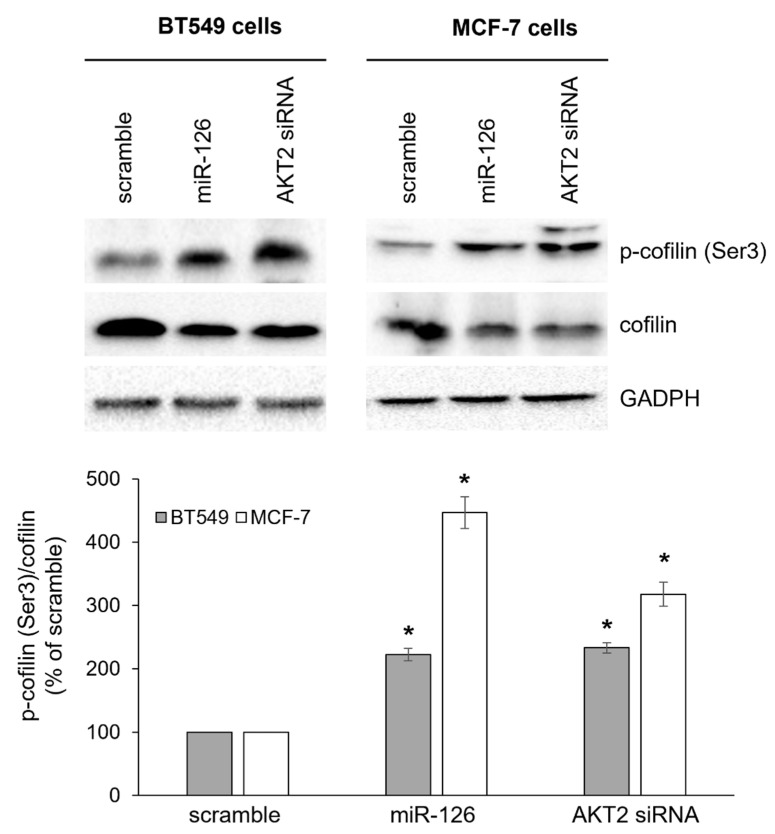
miR-126-3p/AKT2 axis decreases cofilin activity. BT549 and MCF-7 cells were left untreated (NT) or transiently transfected with scramble oligo (scramble), miR-126-3p mimic (miR-126), or AKT2 siRNA before investigating the expression of total and phosphorylated (Ser3) cofilin by Western blot. Histogram shows densitometric analysis; values are reported as p-cofilin (Ser3)/total cofilin ratio compared to scramble-transfected cells (not statistically differing from NT cells), arbitrarily set to 100%. GAPDH was used as loading control. Blots are representative of five independent experiments and results are reported as mean ± SEM. *: *p* < 0.01 versus scramble.

**Figure 5 ijms-23-05484-f005:**
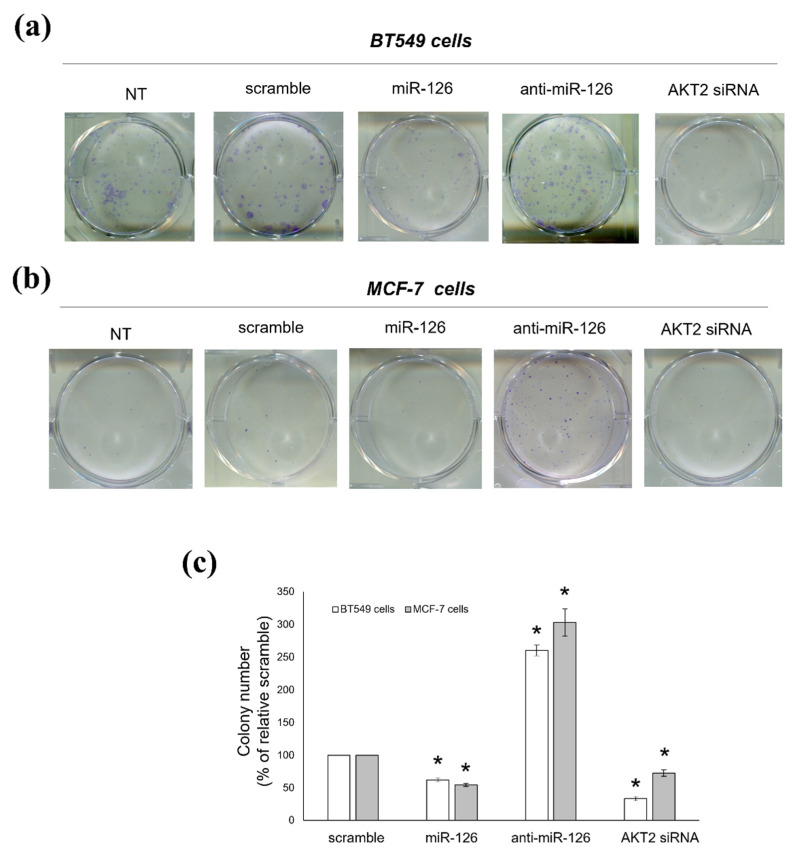
miR-126-3p/AKT2 axis reduces the clonogenic potential of BC cells. Colony forming unit (CFU) assay performed with BT549 (**a**) and MCF-7 (**b**) cells left untreated (NT) or transiently transfected with either scramble oligo (scramble), miR-126-3p-mimic (miR-126), miR-126-3p-inhibitor (anti-miR-126), or AKT2 siRNA. Photographs are representative of three independent experiments. (**c**) Histograms show the colony number reported as percentage of scramble-transfected cells (not statistically differing from NT cells), arbitrarily set to 100% (absolute colony number = 94.67 ± 2.21 and 33.12 ± 0.18 for BT549 and MCF-7 cells, respectively). Data are shown as mean ± SEM. *: *p* < 0.01 versus relative scramble.

**Figure 6 ijms-23-05484-f006:**
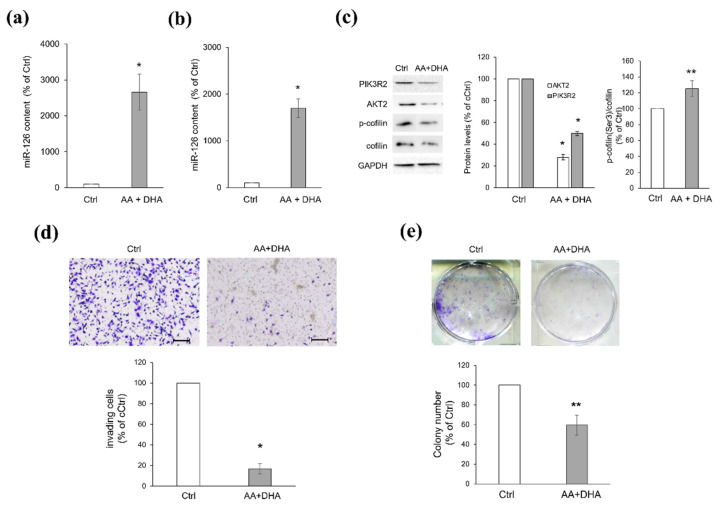
miR-126-3p-enriched MVs derived from PUFA-treated platelets decrease tumorigenic properties of BT549 cells by directly targeting AKT2. (**a**) RT-PCR analysis of *miR-126* content in MVs derived from resting (Ctrl) or AA plus DHA-treated (AA + DHA) platelets and (**b**) in BT549 cells after MV delivery. Histograms show *miR-126* expression reported as percentage of relative Ctrl, arbitrarily set to 100%. (**c**) Western blot analysis of PIK3R2, AKT2, and phosphorylated cofilin levels in BT549 cells treated as in (**b**). Blots are representative of four independent experiments. Histograms represent the densitometric analysis, expressed as percentage of Ctrl, arbitrarily set to 100%. (**d**) Transwell invasion assay performed with BT549 cells treated as in (**b**). Images are representative of three independent experiments. Histogram shows the number of invading cells, reported as percentage of Ctrl, arbitrarily set to 100% (absolute cell number = 284 ± 17.24). (**e**) CFU assay performed with BT549 cells treated as in (**b**). Photographs are representative of three independent experiments. Histograms show colony number reported as percentage of Ctrl arbitrarily set to 100% (absolute colony number = 162.27 ± 4.3). Data in histograms are shown as mean ± SEM. * *p* < 0.01 and ** *p* < 0.05 versus relative Ctrl. Scale bars: 250 µm.

**Figure 7 ijms-23-05484-f007:**
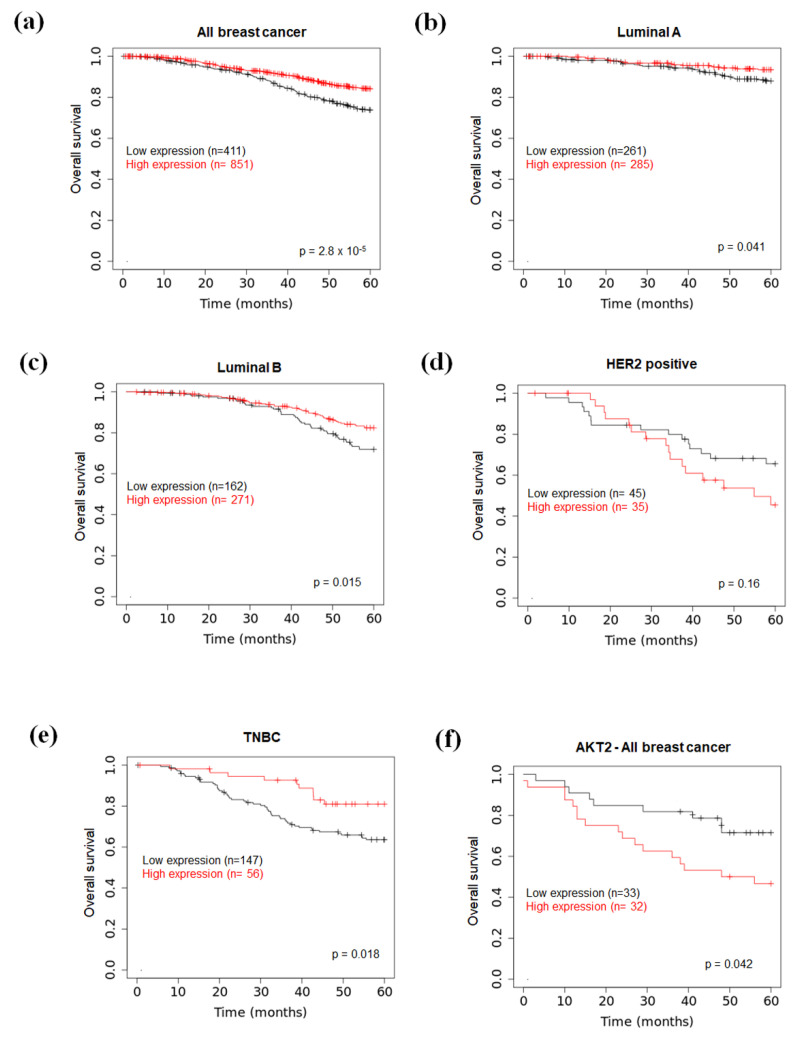
Kaplan–Meier survival analysis of BC patients based on miR-126 expression and AKT2 protein levels. Graphs were downloaded from the KM plotter website (https://kmplot.com/analysis/, accessed on 26 December 2021) [52]. Comparison of 5-year overall survival between patients with high and low miR-126 expression levels in (**a**) all breast cancers, (**b**) luminal A, (**c**) luminal B, (**d**) HER2 positive, and (**e**) TNBC tumors. Data were from METABRIC database. (**f**) Comparison of 5-year overall survival between BC patients with high and low AKT2 protein levels. Data were from [53].

**Figure 8 ijms-23-05484-f008:**
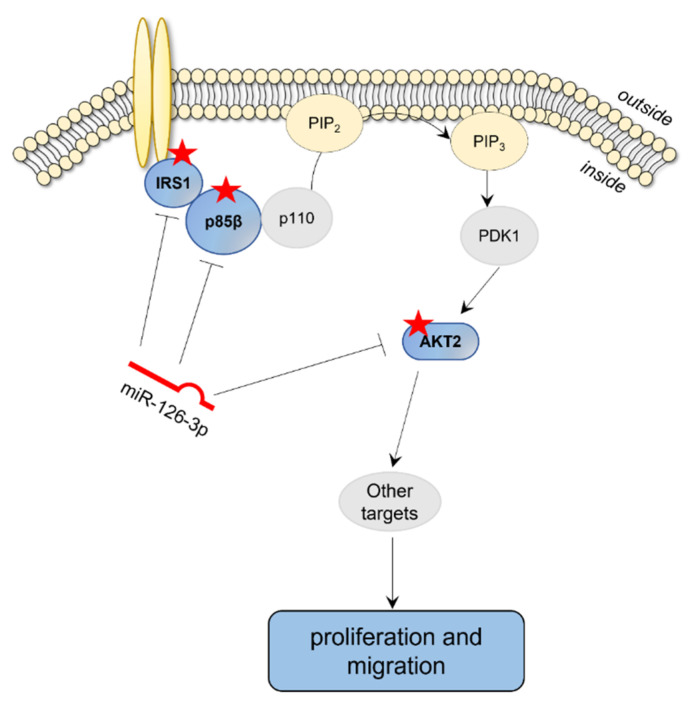
Schematic representation of miR-126-3p regulation of PI3K/AKT signaling and effects on BC cells. Red stars: miR-126-3p direct targets.

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
