# Peer review of "Platelet-Derived miR-126-3p Directly Targets AKT2 and Exerts Anti-Tumor Effects in Breast Cancer Cells: Further Insights in Platelet-Cancer Interplay"

_ijms, 2022, doi:10.3390/ijms23105484_

Round 1

Reviewer 1 Report

The present article by Sibilano et al. describes the anti-cancer effects of platelet-derived microvesicles. The relevance of this material is not in doubt, but there are a number of minor comments that can improve its quality:
1. It is desirable to add to all experiments the data obtained on the control cell lines BT-549 and MCF-7
2. Add data on cell migration to the MCF-7 line
3. Data on the change in cofilin activity on the MCF-7 cell line should be added.

Author Response

First, we would like to thank the constructive suggestions made by Referee, with helpful criticisms that improved the clarity of our work. A point-by-point reply to the criticism raised is detailed below.

The present article by Sibilano et al. describes the anti-cancer effects of platelet-derived microvesicles. The relevance of this material is not in doubt, but there are a number of minor comments that can improve its quality:

  1. It is desirable to add to all experiments the data obtained on the control cell lines BT-549 and MCF-7

As expected, no statistically significant differences were observed between not-transfected (NT) and scramble-transfected cells. As kindly suggested by the Referee, we added panels of NT cells in each figure but, for sake of clarity, calculations were made by using scramble-transfected cells as reference control (as indicated in Figure legends).

  1. Add data on cell migration to the MCF-7 line

Data on cell migration of MCF-7 cells are identical to those obtained with the BT549 cell line. However, in this case, migration was not assessed by Incucyte live-imaging, but with manual scratch and visualization by light microscope. Therefore, we decided to put results concerning MCF-7 cells as data not shown. For your convenience, you can see the MCF-7 cell migration in “Figure 1 for Referee”.

  1. Data on the change in cofilin activity on the MCF-7 cell line should be added.

We put data on cofilin activity of MCF-7 cells in the revised version of the manuscript.

We hope the current version is satisfying.

Reviewer 2 Report

Major Comments

-The title should be more declarative, right now it’s very bland and as a reader I wouldn’t be drawn to reading it. Eg ‘Platelet-derived microvesicles containing Mir-126-3p target AKT2 and suppress breast cancer cell growth’ or something like that.

- If mir-126-3p is the actual miRNA downregulating AKT2, then it should be referred to as such in all Figures and text. This reports more accurately the phenonema and adds accuracy to the literature. All references to AKT2 as a gene or mRNA should be italicised. This also includes uses of mir-126-3p and anyone gene referred to in the manuscript eg EGFL7, RPL21 etc.

-Please show evidence that AKT2 3’UTR mir-126-3p target site is conserved or not (PhastCon or alignment) and discuss. This is important for reporting this result to the literature.

-Figure 1c: reorder graph and Western and put AKT2 first

-Figure 2: there is no mention of the red dashed lines and there are no scale bars. The columns of photos should be labelled with 0 and 15-18 hr

-Figure 4: cofilin is not mentioned in y-axis label, swap total cofilin with p-cofilin in Western. All references to p-cofilin should be pS3-cofilin.

-Figure 6a & b: these qRT-PCR don’t look normalised to control. Perhaps try log10 scale. What is the reference gene/RNA here? Please remove the Ct curves or put in supplementary.

-Figure 6c: p-cofilin – is this normalised? This results is not very believable that with a ~20% increase in p-cofilin this is p<0.01 where PIK3R2 and AKT2 after AA+DHA treatment are 50% decreased and p-value is only <0.05. Please explain? And/or remove.

-Figure 7: what FPKM expression level was used as the median to stratify low and high expression here? The overall survival differences are very subtle.

-There are frequent uses of the term ‘expression’ and ‘up-regulation’ during when referring to mir-126-3p being added to the cells or via platelet MV delivery. These terms are wrongly used here as there is no expression or regulation occurring here. Please correct the many uses of this eg Line 126, Line 166 (inhibitor does not down-regulate, it blocks), line 211, line 288.

-A summary figure or part of figure showing mir-126-3p regulation of AKT2 and other members of the PI3/AKT signalling pathway would useful and would increase the readability and application of the work.

Minor Comments

-Line 34: ‘strictly influence’?? please revise.

-Just an example is used multiple times, use ‘As an example,’

-Line 56: ‘discrepancies’, replace with phenomena.

- Line 62: ‘This miRNA, encoded by has-mir-126, ‘

- Line 79: put citation at end of the sentence

- Line 91: ‘two’ this doesn’t meld well with above sentences describing AKT2 isoforms.

- Line 108: ‘highly enhanced’ – redundant usage.

- Line 109: go=gain.

- Figure 1 legend: ‘Transfected or ectopic’ mir-126-3p downregulates AKT2

- Line 145: ‘gene interfering’ ???

- Line 149: ‘superimposable=identical’

- Figure 1b ‘Luciferasi’??

- Line 166: coherently??

- Figures 3 & 6 – put scale bar numbers in legend as they are too hard to see.

- Figure 3 legend. ‘The mir-126-3p/….’

- Line 193: Noteworthy=Notably

- Line 230: Figure 6b

- Line 384: Targetscan 8.0, 5.1 mentioned in Figure 1.
- 4.6 AKT2 3’UTR

- Line 425: Sentence should not start with a number, ‘BT549 (1.0x10^4) cells

- Line 465: The normalisation to RPL21 has not been made clear in Figure 6a & b.

Author Response

POINT TO POINT REPLY

First, we would like to thank the constructive suggestions made by Referee, with helpful criticisms that improved the clarity of our work. A point-by-point reply to the criticism raised is detailed below.

Major Comments

1) The title should be more declarative, right now it’s very bland and as a reader I wouldn’t be drawn to reading it. Eg ‘Platelet-derived microvesicles containing Mir-126-3p target AKT2 and suppress breast cancer cell growth’ or something like that.

We thank the referee for his/her suggestion. We agree in using a title more declarative and now it has been changed accordingly. We hope the current title is satisfying.

2) If mir-126-3p is the actual miRNA downregulating AKT2, then it should be referred to as such in all Figures and text. This reports more accurately the phenonema and adds accuracy to the literature. All references to AKT2 as a gene or mRNA should be italicised. This also includes uses of mir-126-3p and anyone gene referred to in the manuscript eg EGFL7, RPL21 etc.

As kindly suggested, we specified mir-126-3p throughout the text and we also put it in the Figures, unless referring to precursor miR-126. We also italicized all genes and mRNAs in the manuscript.

3) Please show evidence that AKT2 3’UTR mir-126-3p target site is conserved or not (PhastCon or alignment) and discuss. This is important for reporting this result to the literature.

We completely agree with Referee’s comment. Conservation of AKT2 3’UTR mir-126-3p target sequence (as taken from TargetScan website) is shown in “Figure 2 for Referee”; we preferred not to insert it in the paper, so as not to weigh down the reading.

4) Figure 1c: reorder graph and Western and put AKT2 first

We thank the Referee. We reordered western and graph. Now AKT2 is displayed first.

5) Figure 2: there is no mention of the red dashed lines and there are no scale bars. The columns of photos should be labelled with 0 and 15-18 hr.

We apologize for the negligence. In the new Figure, we put scale bars and time labelling in the columns of photos. Mention of the red dashed lines is present in the legend of Figure 2.

6) Figure 4: cofilin is not mentioned in y-axis label, swap total cofilin with p-cofilin in Western. All references to p-cofilin should be pS3-cofilin.

We apologize for this clerical error. We corrected y-axis legend. Now, p-cofilin is also displayed first. Finally, p-cofilin (Ser3) is mentioned everywhere.

7) Figure 6a & b: these qRT-PCR don’t look normalised to control. Perhaps try log10 scale. What is the reference gene/RNA here? Please remove the Ct curves or put in supplementary.

We apologize for the omission, as we selected only the absolute curves for miR-126. Obviously, amounts of miR-126 have been normalized against the housekeeping RPL21 gene (as indicated in the Materials and Methods section). We preferred to remove the Ct curves.

8) Figure 6c: p-cofilin – is this normalised? This results is not very believable that with a ~20% increase in p-cofilin this is p<0.01 where PIK3R2 and AKT2 after AA+DHA treatment are 50% decreased and p-value is only <0.05. Please explain? And/or remove.

We think there was a misunderstanding, as statistical significance is p<0.01 for PIK3R2 and AKT2 and p<0.05 for p-cofilin; this is clearly in line with percentages of inhibition.

9) Figure 7: what FPKM expression level was used as the median to stratify low and high expression here? The overall survival differences are very subtle.

Patients were split by setting “auto select best cut-off option” that uses, as cutoff, the best performing threshold. By this way, the cutoff value, used in the analysis, was 9.33; expression range of the probe was 6-13.

10) There are frequent uses of the term ‘expression’ and ‘up-regulation’ during when referring to mir-126-3p being added to the cells or via platelet MV delivery. These terms are wrongly used here as there is no expression or regulation occurring here. Please correct the many uses of this eg Line 126, Line 166 (inhibitor does not down-regulate, it blocks), line 211, line 288.

As kindly suggested, we modified sentences accordingly.

11) A summary figure or part of figure showing mir-126-3p regulation of AKT2 and other members of the PI3/AKT signalling pathway would useful and would increase the readability and application of the work.

We thank the Referee for his/her suggestion. In the revised version of the manuscript, we added the new Figure 8 summarizing all demonstrated effects of miR-126-3p on PI3K/AKT signaling, to increase readability and application of our work. We hope the Referee will appreciate this schematic picture.

Minor Comments

1) Line 34: ‘strictly influence’?? please revise.

2) Just an example is used multiple times, use ‘As an example,’

3) Line 56: ‘discrepancies’, replace with phenomena.

4)  Line 62: ‘This miRNA, encoded by has-mir-126, ‘

As kindly suggested, we modified all the sentences.

5) Line 79: put citation at end of the sentence

We apologize for the error. We put citation at end of the sentence, according to the Referee’s suggestion.

6) Line 91: ‘two’ this doesn’t meld well with above sentences describing AKT2 isoforms.

We apologize for the misunderstanding. We were referring to different AKT isoforms and not AKT2 isoforms. To make reading easier and smoother, we modified the sentence.

7) Line 108: ‘highly enhanced’ – redundant usage.

8) Line 109: go=gain.

9) Figure 1 legend: ‘Transfected or ectopic’ mir-126-3p downregulates AKT2

We revised the sentences, according to the Referee’s suggestion.

10) Line 145: ‘gene interfering’

We apologize for the negligence. As siRNA is a small interfering RNA that targets (and blocks) messenger RNA, we corrected text accordingly.

11) Line 149: ‘superimposable=identical’

12) Figure 1b ‘Luciferasi’??

13) Line 166: coherently??

We revised sentences, according to the Referee’s suggestion.

14) Figures 3 & 6 – put scale bar numbers in legend as they are too hard to see.

We put scale bar numbers in the legend of Figures, according to the Referee’s suggestion.

15) Figure 3 legend. ‘The mir-126-3p/….’

16) Line 193: Noteworthy=Notably

17) Line 230: Figure 6b

We modified sentences, as requested.

18) Line 384: Targetscan 8.0, 5.1 mentioned in Figure 1.

We apologize for the mistake. Now, TargetScan 8.0 is correctly mentioned in the text.

19) 4.6 AKT2 3’UTR

20) Line 425: Sentence should not start with a number, ‘BT549 (1.0x10^4) cells

We revised the sentences, according to the Referee’s suggestion.

21) Line 465: The normalisation to RPL21 has not been made clear in Figure 6a & b.

Please, see comments to Point 7 in the “Major comments” section.
